# Impact of Adjuvant Chemotherapy on Survival of Patients with Advanced Residual Disease at Radical Cystectomy following Neoadjuvant Chemotherapy: Systematic Review and Meta-Analysis

**DOI:** 10.3390/jcm10040651

**Published:** 2021-02-08

**Authors:** Wojciech Krajewski, Łukasz Nowak, Marco Moschini, Sławomir Poletajew, Joanna Chorbińska, Andrea Necchi, Francesco Montorsi, Alberto Briganti, Rafael Sanchez-Salas, Shahrokh F Shariat, Juan Palou, Marek Babjuk, Jeremy YC Teoh, Francesco Soria, Benjamin Pradere, Paola Irene Ornaghi, Aleksandra Pawlak, Janusz Dembowski, Romuald Zdrojowy

**Affiliations:** 1Department of Urology and Oncologic Urology, Wrocław Medical University, 50-556 Wroclaw, Poland; lllukasz.nowak@gmail.com (Ł.N.); joanna.chorbinska@gmail.com (J.C.); janusz.dembowski@umed.wroc.pl (J.D.); romuald.zdrojowy@umed.wroc.pl (R.Z.); 2Klinik für Urologie, Luzerner Kantonsspital, 6004 Lucerne, Switzerland; marco.moschini87@gmail.com (M.M.); paolairene.ornaghi@gmail.com (P.I.O.); 3Second Department of Urology, Centre of Postgraduate Medical Education, 01-813 Warsaw, Poland; slawomir.poletajew@cmkp.edu.pl; 4Department of Medical Oncology, Fondazione IRCCS Istituto Nazionale dei Tumori, 20133 Milan, Italy; andrea.necchi@istitutotumori.mi.it; 5Unit of Urology, Urological Research Institute (URI), IRCCS Ospedale San Raffaele, 20132 Milan, Italy; montorsi.francesco@unisr.it (F.M.); briganti.alberto@hsr.it (A.B.); 6Department of Urology, Institute Mutualiste Montsouris, Université Paris-Descartes, 75014 Paris, France; rafael.sanchez-salas@imm.fr; 7Department of Urology, Medical University of Vienna, 1090 Vienna, Austria; shahrokh.shariat@meduniwien.ac.at (S.F.S.); benjaminpradere@gmail.com (B.P.); 8Department of Urology, University of Texas Southwestern Medical Center, Dallas, TX 75390, USA; 9Departments of Urology, Weill Cornell Medical College, New York, NY 10065, USA; 10Institute for Urology and Reproductive Health, I.M. Sechenov First Moscow State Medical University, 119146 Moscow, Russia; 11Division of Urology, Department of Special Surgery, Jordan University Hospital, The University of Jordan, Amman 11942, Jordan; 12European Association of Urology Research Foundation, 6803 AA Arnhem, The Netherlands; 13Department of Urology, Second Faculty of Medicine and Hospital Motol, Charles University, 15006 Prague, Czech Republic; Marek.Babjuk@fnmotol.cz; 14Fundació Puigvert, Department of Urology, Autonomous University of Barcelona, 08025 Barcelona, Spain; jpalou@fundacio-puigvert.es; 15S.H. Ho Urology Centre, Department of Surgery, Prince of Wales Hospital, The Chinese University of Hong Kong, Hong Kong; jeremyteoh@surgery.cuhk.edu.hk; 16Division of Urology, Department of Surgical Sciences, San Giovanni Battista Hospital, University of Studies of Torino, 10124 Turin, Italy; soria.fra@gmail.com; 17Department of Oncology and Urology, University Hospital of Tours, 37000 Tours, France; 18Department of Pharmacology and Toxicology, Faculty of Veterinary Medicine, Wroclaw University of Environmental and Life Sciences, 50-375 Wrocław, Poland; aleksandra.pawlak@upwr.edu.pl

**Keywords:** adjuvant chemotherapy, muscle-invasive bladder cancer, neoadjuvant chemotherapy

## Abstract

Background: Cisplatin-based neoadjuvant chemotherapy (NAC) followed by radical cystectomy (RC) with pelvic lymph-node dissection is the standard treatment for cT2-4a cN0 cM0 muscle-invasive bladder cancer (MIBC). Despite the significant improvement of primary-tumor downstaging with NAC, up to 50% of patients are eventually found to have advanced residual disease (pT3–T4 and/or histopathologically confirmed nodal metastases (pN+)) at RC. Currently, there is no established standard of care in such cases. The aim of this systematic review and meta-analysis was to assess differences in survival rates between patients with pT3–T4 and/or pN+ MIBC who received NAC and surgery followed by adjuvant chemotherapy (AC), and patients without AC. Materials and Methods: A systematic search was conducted in accordance with the PRISMA statement using the Medline, Embase, and Cochrane Library databases. The last search was performed on 12 November 2020. The primary end point was overall survival (OS) and the secondary end point was disease-specific survival (DSS). Results: We identified 2124 articles, of which 6 were selected for qualitative and quantitative analyses. Of a total of 3096 participants in the included articles, 2355 (76.1%) were in the surveillance group and 741 (23.9%) received AC. The use of AC was associated with significantly better OS (hazard ratio (HR) 0.84, 95% confidence interval (CI) 0.75–0.94; *p* = 0.002) and DSS (HR 0.56, 95% CI 0.32–0.99; *p* = 0.05). Contrary to the main analysis, in the subgroup analysis including only patients with pN+, AC was not significantly associated with better OS compared to the surveillance group (HR 0.89, 95% CI 0.58–1.35; *p* = 0.58). Conclusions: The administration of AC in patients with MIBC and pT3–T4 residual disease after NAC might have a positive impact on OS and DSS. However, this may not apply to N+ patients.

## 1. Introduction

According to European Association of Urology (EAU) guidelines, cisplatin-based neoadjuvant chemotherapy (NAC) followed by radical cystectomy (RC) with bilateral pelvic lymph-node dissection is the standard treatment of cT2-4a cN0 cM0 muscle-invasive bladder cancer (MIBC) [1]. NAC administration is consequently associated with an 8% absolute improvement in five-year overall survival (OS) [2]. However, despite the significant effect of primary-tumor downstaging with NAC, up to 50% of patients are eventually found to have advanced residual disease (pT3–T4 and/or histopathologically confirmed nodal metastases (pN+)) at RC [3,4]. Currently, there is no established standard of care in such cases.

Although several studies and meta-analyses investigated whether post-RC adjuvant chemotherapy (AC) in chemotherapy-naïve patients with pT3–T4 and/or pN+ disease can improve oncological outcomes [5,6], data regarding the role of AC after NAC and RC are scarce.

The aim of this systematic review and meta-analysis was to assess differences in survival rate between patients with pT3–T4 and/or pN+ MIBC who received NAC and surgery followed by AC, and patients without AC.

## 2. Materials and Methods

### 2.1. Search Strategy

The systematic review and meta-analysis were performed in accordance with the Preferred Reporting Items for Systematic Reviews and Meta-Analyses (PRISMA) statement and the Cochrane Handbook for Systematic Reviews of Interventions [7,8]. Study protocol was established in priori and was registered with PROSPERO (CRD42021226369).

Two review authors (L.N. and W.K.) independently conducted a systematic search through three electronic databases: Medline, Embase, and Cochrane Library. The last search was performed on 12 November 2020. The following terms and/or keywords were used: (“adjuvant chemotherapy” OR “AC”) AND (“neoadjuvant chemotherapy” OR “NAC”) AND (“bladder cancer” OR “muscle-invasive bladder cancer” OR “MIBC”) AND (“residual disease” OR “locally advanced disease”). No specific time or language restrictions were applied. A cross-reference search was also performed on articles selected for full-text review. Additional articles were screened from articles published ahead of print in various urological journals.

### 2.2. Inclusion Criteria

Studies were included if they met all of the following criteria: (1) comparing patients with pT3–T4 and/or pN+ disease at RC who received NAC and surgery followed by AC, with those without AC; (2) reporting at least one survival outcome of interest, including overall survival (OS) and disease-specific survival (DSS); (3) reported median follow-up of a minimum of 12 months; and (4) retrospective or prospective design. Reviews, case reports, letters, or commentaries were excluded.

The primary end point of this systematic review and meta-analysis was OS, and the secondary end point was DSS.

### 2.3. Data Extraction

After removal of duplicates, two review authors (W.K. and L.N.) independently screened the titles and abstracts of the retrieved records using a standardized item form. All potentially eligible studies were evaluated as full text if available. Any disagreements were subsequently resolved by consultation with the other authors.

The following data were initially extracted: first author, year of publication, journal, study region, study design, study duration, number of patients in AC and surveillance group, and median follow-up. Further, the following clinicopathological data were retrieved: age, gender, Charlson comorbidity index (CCI), NAC regimen, number of NAC cycles, pathological stage, positive nodal status, surgical margin status, AC regimen, and number of AC cycles.

Subsequently, we extracted the outcome measurements of OS and DSS, including the hazard ratios (HRs) and 95% confidence intervals (CIs). OS was primarily defined as time from the date of RC to death from any case or to the time of the last known follow-up. DSS was primarily defined as the time from RC to death from cancer/MIBC or to the time of the last known follow-up.

For articles that lacked some data, the corresponding authors were contacted, requesting additional information from their research, but no additional data were received.

### 2.4. Quality and Risk-of-Bias Assessment

The quality of the selected studies was assessed independently by two review authors (W.K. and L.N.). The evaluation of the methodological quality of eligible studies was performed according to the Newcastle–Ottawa Scale (NOS) [9].

Risk-of-bias (RoB) was determined using the pragmatic approach for the evaluation of nonrandomized studies by examining the adjustments for confounders according to the Cochrane Handbook for Systematic Reviews of Interventions [8]. To this end, the articles were reviewed on the basis of the adjustment for the effects of age, gender, CCI, pathological stage, positive surgical margin, and type of NAC regimen. RoB and confounder evaluation in each study were independently assessed by two authors (W.K. and M.M.), and disagreements were resolved by consultation with the other authors. To assess the publication bias of the selected studies, funnel plots were generated and evaluated for asymmetry.

### 2.5. Statistical Analysis

Statistical analysis was conducted using Review Manager 5.3 (The Nordic Cochrane Center, The Cochrane Collaboration, Copenhagen, Denmark) and the R platform (R project, Vienna, Austria). If available, the reported HRs and 95% CIs were included in the meta-analysis. We preferred to collect multivariate-analysis data; otherwise (if not reported), data from univariate analyses were extracted. The statistical significance of the pooled HRs was evaluated by the *Z* test. The heterogeneity I^2^ index was calculated in order to indicate the proportion of inconsistency between studies that could not be attributed to chance. Significant heterogeneity was indicated by a ratio >50% for I^2^ or a *p* value ≤0.10 in the Cochrane Q test, in which case a random-effect model was used. When no significant heterogeneity was observed among the studies, a fixed-effect model was used. For all other tests, *p* ≤ 0.05 was considered a statistically significant difference.

## 3. Results

### 3.1. Study Identification and Quality Assessment

Our search strategy initially identified 2124 articles (2098 from online databases and 26 from additional sources). A flow diagram of the study selection and subsequent exclusions (with reasons) is presented in Figure 1. Eventually, we identified six studies for qualitative and quantitative analyses [10,11,12,13,14,15]. Table 1 summarizes the baseline characteristics of the eligible studies. Of the 3096 total participants in the selected articles, 2355 (76.1%) and 741 (23.9%) were in the surveillance and AC groups, respectively. All included studies had a retrospective design. The reported median follow-up ranged from 30 to 50 months. The assessment of the quality scores for the selected trials based on NOS ranged from 5 to 7, which was considered adequate for the subsequent systematic review and meta-analysis.

### 3.2. Clinical Characteristics and Pathological Data

Table 2 summarizes the clinicopathological characteristics of patients in the AC and surveillance groups. Generally, no statistically significant differences in the baseline clinical parameters (age, gender, and CCI) were observed in the specific studies. Patients receiving AC were significantly younger in two studies [10,11]. Detailed data regarding the NAC regimen and number of administered NAC cycles were reported in three out of the six articles [10,11,15]. The majority of patients in both the AC and surveillance groups received cisplatin-based NAC regimens. The median number of NAC cycles ranged from 3 to 5. One study only reported data for patients with pN+ [10]. In the five other trials, rates of pN+ in the AC and surveillance groups ranged from 45.5% to 83% and 45.8% to 50%, respectively [11,12,13,14]. The surgical margin status was reported in three of the included articles, and positive rates ranged from 17% to 24% and 16% to 34% in the AC and surveillance groups, respectively [11,13,15]. Only one study provided data regarding the proportion of patients with pure variant histology (VH)—21.7% of patients had pure VH in both groups [11]. Detailed data regarding the AC regimen and the number of administered AC cycles were reported in three of the six studies [10,11,15]. In two trials, the majority of patients received cisplatin-based AC regimens [10,11], while in one study, a carboplatin-based AC regimen was predominantly used [15]. The median number of AC cycles ranged from four to five.

### 3.3. Risk-of-Bias Assessment

All studies carried a high RoB, which was primarily related to their retrospective design. In the majority of studies, multivariate analyses were adjusted for the effect of at least three major confounders, with age, gender, and CCI being the most common.

### 3.4. Meta-Analysis Results

For each outcome of interest (OS and DSS), we performed a main analysis that included data from all available publications. Subsequently, we conducted subgroup analysis of OS in the pN+ subpopulation. Subgroup analysis of DSS in the pN+ subpopulation could not be reliably performed due to insufficient data.

OS data were reported in five articles [10,11,12,13,14]. No significant heterogeneity was observed among the studies (I^2^ = 0%; *p* = 0.61). Therefore, a fixed-effect model was used. A forest plot of HR and 95% CI for OS is presented in Figure 2A. The use of AC was associated with significantly better OS (HR 0.84, 95% CI 0.75–0.94; *p* = 0.002) in the pooled analysis. The funnel plot revealed no publication bias (Figure 3A).

Data for DSS were reported in two studies [10,15]. No significant heterogeneity was observed among the studies (I^2^ = 0%; *p* = 0.35). Therefore, a fixed-effect model was used. A forest plot of HR and 95% CI for OS is presented in Figure 2B. Receiving AC was associated with significantly better DSS (HR 0.56, 95% CI 0.32–0.99; *p* = 0.05) in the pooled analysis. The funnel plot revealed no publication bias (Figure 3B).

In subgroup analysis, which only included patients with pN+, AC was not significantly associated with better OS compared to the surveillance group (HR 0.89, 95% CI 0.58–1.35; *p* = 0.58) (Figure 4). Significant heterogeneity was observed among the studies (I^2^ = 59%; *p* = 0.08). Therefore, a random-effect model was used.

## 4. Discussion

In this systematic review and meta-analysis, we provide a summary of the accumulated evidence in a tentatively understood area of MIBC management. We evaluated the impact of AC administration on the survival outcomes of patients with advanced disease (pT3–T4 and/or pN+) at RC after NAC. To the best of our knowledge, this is the first meta-analysis to address this issue. Our analyses demonstrated that AC was associated with significantly better OS (HR 0.84, 95% CI 0.75–0.94; *p* = 0.002) and DSS (HR 0.56, 95% CI 0.32–0.99; *p* = 0.05) compared to the surveillance approach.

The available literature regarding the role of AC in patients with MIBC who had received NAC and demonstrated advanced disease (pT3–T4 and/or pN+) comprises several retrospective studies and population-based registries, whereas no prospective RCT has been published to date. Initial results (only abstract available) from the Retrospective International Study of Cancers of the Urothelium (RISC) group showed that AC administration in patients with residual disease after NAC was associated with a significant reduction in relapse risk compared to surveillance (HR = 0.35 for AC, 95% CI 0.17–0.74). The subset analysis of patients with residual pathologic T4 and/or N+ after NAC also revealed a significant improvement in time to relapse in the AC group (HR = 0.43 for AC, 95% CI 0.21–0.89). Therefore, the authors concluded that AC might delay recurrence in patients with residual disease after NAC in an advanced MIBC setting. Notably, the majority of the included patients changed regimens between NAC and AC [16]. Previously, Millikan et al. suggested that, with the absence of a pathological response to the initial NAC, further therapy with the same regimen may not afford additional survival benefits. Their conclusions were based on the observation that patients either receiving two courses of neoadjuvant MVAC (a chemotherapy combination that includes methotrexate, vinblastine, doxorubicin and cisplatin) followed by RC and three additional cycles of chemotherapy, or undergoing an initial RC followed by five cycles of AC, exhibited comparable progression-free survival (PFS) and DSS [17]. These highlighted reports suggest that, in the case of the absence of a pathological response after NAC, changes in therapeutic regimens should be considered if further AC is subsequently planned. Among the six studies included in this systematic review and meta-analysis, only one provided detailed data regarding changes between pre- and post-RC chemotherapy. In a study by Zargar-Shoshtari et al., 86% of patients received different chemotherapy combinations before and after surgery, but AC was not associated with significant improvements in either RFS or DSS compared to the surveillance approach. A clinically noteworthy observation was made with respect to patients who had been treated with cisplatin-based AC. In these patients, the time to recurrence was the longest when compared with the two other AC regimens (i.e., adjuvant carboplatin and other regimens)—23 vs. 8.4 vs. 5.2, respectively. However, the number of cases in this sub-analysis was small, and differences were not statistically representative [15].

We performed an additional subgroup analysis for OS, only including data from studies reporting HRs for the population with pN+ disease. Contrary to the main analysis, we found that AC was not significantly associated with better OS compared to surveillance (HR 0.89, 95% CI 0.58–1.35; *p* = 0.58) in the pooled analysis. The observed heterogeneity between the studies might be primarily related to the different sizes of the populations. In one included study, Parker et al. provided additional subgroup analyses for the pT3 pN0 pM0 and pT4 pN0 pM0 stages, and demonstrated that AC administration after NAC and RC only significantly improved OS in patients with pT4 pN0 pM0 disease (HR 0.56 for AC, 95% CI 0.33–0.97) [12]. Thus, the increased survival benefit of additional AC could theoretically be observed in patients with the highest T-stage tumors without concomitant nodal metastases. However, identification of additional predictive biomarkers moving beyond the pathological stage to personalize treatment and clinical decisions seems to be a reasonable direction for further research.

As checkpoint inhibitors are increasingly utilized for the treatment of metastatic MIBC, the potential decision as to whether AC should be administered in patients with adverse pathology features after NAC may become debatable, taking into consideration that up to 50% of patients after RC are ineligible for cisplatin-containing chemotherapy [1,18]. Currently, trials with checkpoint immunotherapy are specifically targeting patients at a high risk of recurrence, including patients with significant residual disease after prior NAC (pT2–T4 and/or pN+; NCT02632409, NCT02450331). Comparison of these two treatment modalities (AC vs. checkpoint inhibitors) in patients receiving NAC could identify the best form of treatment in terms of oncological efficacy.

The important question remains as to whether survival benefits surpass the possible significant negative effect of AC after NAC on patient-reported outcomes. According to a study by Zhang et al. with chemotherapy-naïve patients, self-reported anxiety and depression were increased, while quality of life (QoL) was not decreased in MIBC patients during AC [19]. Data regarding the toxicity and impact on patient-reported outcomes of AC after NAC and RC were not presented in any of the included studies. As there are no additional data available in the literature, this issue should be addressed in future trials.

This study has potential limitations that need to be considered for the interpretation of the results. First, the strength of the conclusions that can be drawn from this meta-analysis is limited by the fact that all included studies were retrospective, with their own limitations, such as selection bias. However, extraction of data from multivariable analyses was possible in almost all eligible articles (mainly adjusted for the effects of major confounders); thus, we were able to minimize bias and establish an acceptable level of comparability. Second, the small sample size of some of the eligible publications may render the results less reliable. Future larger studies may overcome this limitation, providing more robust evidence. Third, we could not fully exclude the bias resulting from possible overlap of some patients in particular articles. Fourth, the absence of detailed data on the NAC and AC regimens and the number of cycles given in some included studies, as well as other important confounders (e.g., VH), potentially limit the conclusions for the current daily practice.

## 5. Conclusions

The administration of AC in patients with MIBC and pT3–T4 residual disease after NAC might have a positive impact on OS and DSS. However, this may not apply to N+ patients. Further prospective studies are required in order to make a definitive statement.

## Figures and Tables

**Figure 1 jcm-10-00651-f001:**
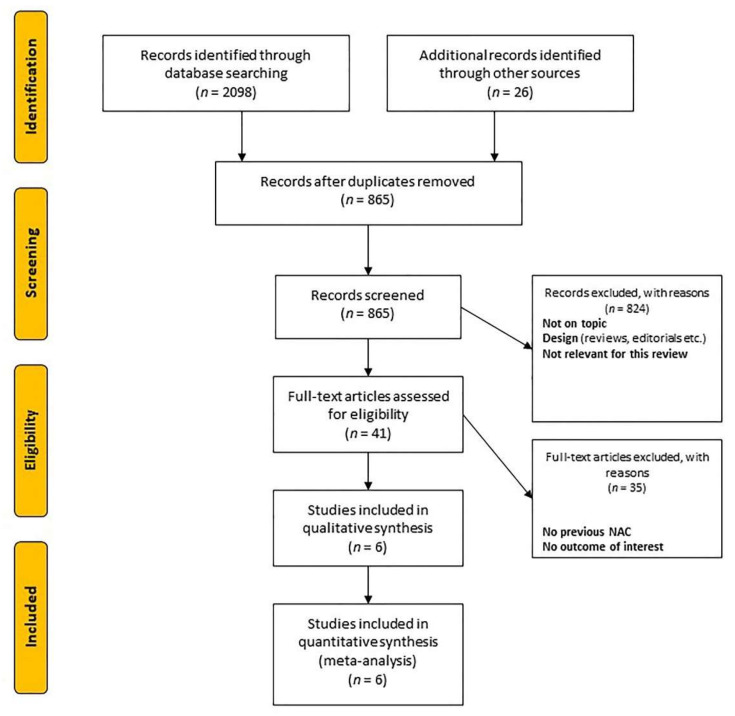
Flow diagram of the study selection.

**Figure 2 jcm-10-00651-f002:**
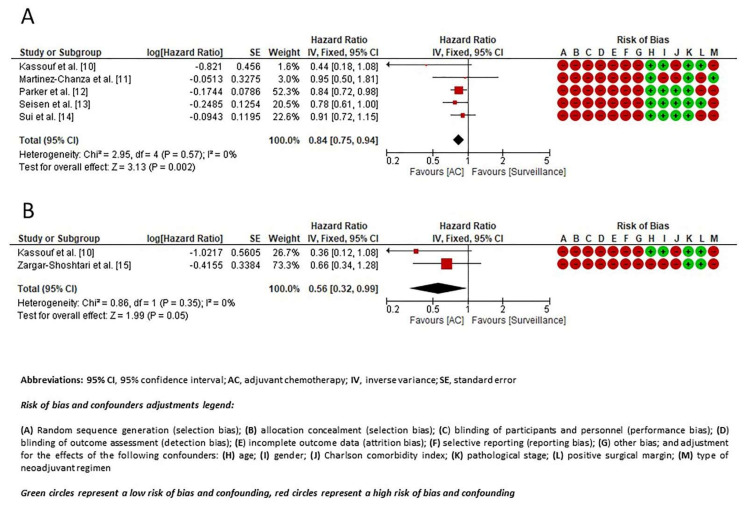
Forest plot of the hazard ratio for (**A**) overall survival, and (**B**) disease-specific survival.

**Figure 3 jcm-10-00651-f003:**
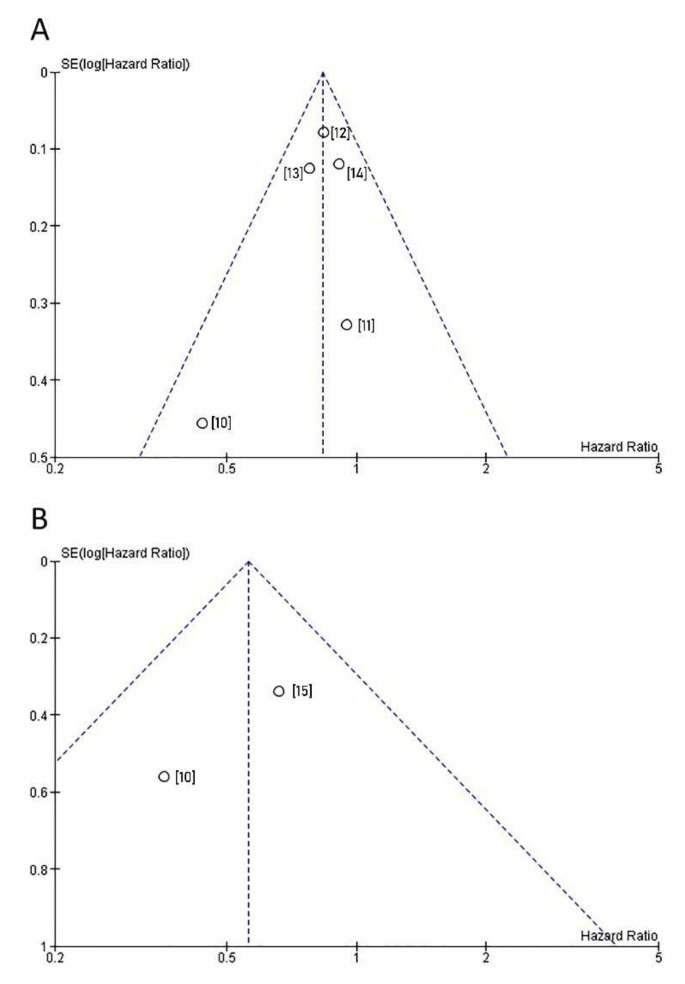
Funnel plot for evaluation of publication bias for (**A**) overall survival, and (**B**) disease-specific survival. Circles represent hazard ratios (x axis) and standard errors (y axis) of particular studies (references are placed in square brackets).

**Figure 4 jcm-10-00651-f004:**
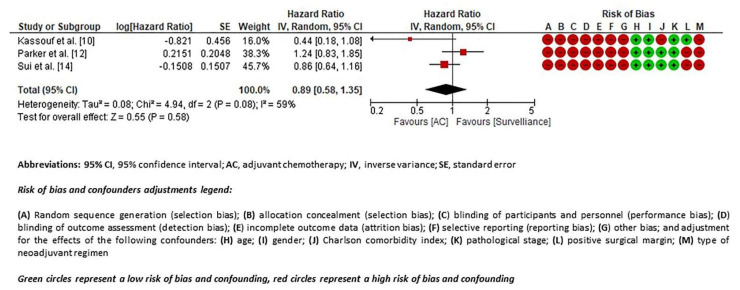
Forest plot of the hazard ratio for overall survival in a subpopulation of pN+ patients.

**Table 1 jcm-10-00651-t001:** Baseline characteristics of the included studies.

Study	Country	Design	Duration	Number of Patients AC/Surveillance	Follow Up Median (Months)	Reported Outcomes of Interest	NOS
[10]	Canada	Retrospective	1993–2003	11/24	50	OS, DSS	5
[11]	United States	Retrospective	1991–2013	23/106	30	OS	7
[12]	United States	Retrospective	2006–2012	326/1033	44.4	OS	6
[13]	United States	Retrospective	2006–2012	184/604	45.7	OS	6
[14]	United States	Retrospective	2004–2013	168/537	44	OS	6
[15]	United States	Retrospective	2001–2013	29/51	NR	DSS	6

AC, adjuvant chemotherapy; DSS, disease-specific survival; NOS, Newcastle–Ottawa Scale; NR, not reported; OS, overall survival.

**Table 2 jcm-10-00651-t002:** Clinicopathological characteristics of the patients in the included studies.

Study	Group	No. of Patients	Age (Median, IQR)	Gender, Male (%)	CCI, (%)	NAC Regimen, (%)	No. of NAC Cycles (Median, IQR)	Clinical Stage (%)	Pathological Stage (%)	Positive Nodal Status (pN+) (%)	Positive Surgical Margins, (%)	VH, (%)	AC Regimen (%)	No. of AC Cycles (Median, IQR)
[10]	AC	11	52	NA	NR	Platinum-based: 81% Other/unknown: 19%	5	NA	NA	100%	NA	NA	Platinum-based: 73% Other/unknown: 27%	NR
	Surveillance	24	64										-	-
[11]	AC	23	61 (51–68)	74%	0: 39% 1–2: 35% >2: 26%	Cisplatin-based: 64% Carboplatin-based: 21% Other/unknown: 13%	3 (3–4)	>cT2: 22% cN+: 17%	pT3a–T4a: 17% pT4b and/or pN+: 83%	NA	17%	21.7%	Cisplatin-based: 47% Carboplatin-based: 22% Other/unknown: 30%	4 (3–4)
	Surveillance	106	66 (59–71)	71%	0: 48% 1–2: 25% >2: 25%	Cisplatin-based: 70% Carboplatin-based: 15% Other/unknown: 15%	3 (3–4)	>cT2: 39% cN+: 16%	pT3a–pT4a: 53% pT4b and/or pN+: 47%	NA	16%	21.7%	-	-
[12]	AC	326	<60 years: 31.8% ≥60 years: 66.8%	73.3%	0: 74.2% 1: 19.6% ≥2: 6.1%	NR	NR	NR	pT3N0: 25.9% pT4N0: 13.7% pTanyN+: 60.4%	60.4%	NR	NR	NR	NR
	Surveillance	1033	<60 years: 32.1% ≥60 years: 67.9%	74.7%	0: 74.4% 1: 20.1% ≥2: 5.4%	NR	NR	NR	pT3N0: 34.8% pT4N0: 11.3% pTanyN+: 53.9%	53.9%	NR	NR	-	-
[13] *	AC	184	mean (SD) 65.7 (10.0)	75.8%	0: 75% 1: 20.3% ≥2: 4.7%	NR	NR	NR	pT3N0: 40.8% pT4N0: 13.7% pTanyN+: 45.5%	45.5%	17.1%	NR	NR	NR
	Surveillance	604	mean (SD) 65.3 (9.3)	76.6%	0: 74.5% 1: 21% ≥2: 5.5%	NR	NR	NR	pT3N0: 39.9% pT4N0: 14.3% pTanyN+: 45.8%	45.8%	16.8%	NR	-	-
[14]	AC	168	<60 years: 38% ≥60 years: 62%	67%	0: 76% 1: 19% ≥2: 5%	NR	NR	NR	≥pT3: 87%	57%	NR	NR	NR	NR
	Surveillance	537	<60 years: 33% ≥60 years: 68%	66%	0: 74% 1: 20% ≥2: 6%	NR	NR	NR	≥pT3: 90%	45%	NR	NR	-	-
[15]	AC	29	66 (56–76)	83%	median (IQR) 3 (2–4)	Cisplatin-based: 79% Carboplatin-based: 21% Other/unknown: 0%	3 (3–4)	>cT2: 48% cN+: 20%	pT3–T4: 76%	83%	24%	NR	Cisplatin-based: 27.6% Carboplatin-based: 55.2% Other/unknown: 17.2%	5 (3–8) 4 (2–6) 4 (4–6)
	Surveillance	51	68 (63–75)	69%	median (IQR) 3 (2–3)	Cisplatin-based: 74% Carboplatin-based: 26% Other/unknown: 0%	3 (3–4)	>cT2: 57% cN+: 29%	pT3–T4: 88%	50%	34%	NR	-	-

* IPTW-weighted study population; abbreviations: AC, adjuvant chemotherapy; CCI, Charlson comorbidity index; cN+, clinically suspected nodal metastases; IQR, interquartile range; IPTW, inverse probability of treatment weighting; NA, not applicable; NAC, neoadjuvant chemotherapy; No., number of; NR, not reported; SD, standard deviation; VH, variant histology.

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
