# Peer review of "Impact of Adjuvant Chemotherapy on Survival of Patients with Advanced Residual Disease at Radical Cystectomy following Neoadjuvant Chemotherapy: Systematic Review and Meta-Analysis"

_jcm, 2021, doi:10.3390/jcm10040651_

Round 1
Reviewer 1 Report
It was my pleasure reviewing "Impact of Adjuvant Chemotherapy on Survival of Patients with Advanced Residual Disease at Radical Cystectomy following Neoadjuvant Chemotherapy: Systematic Review and Meta-Analysis" by Krajewski and colleagues.
The authors present results from 6 studies which were included in the meta-analysis.
However three of the studies : Sui et al, Siesen et al and Parker et al are using the same National Cancer Database and therefore have the same patients. Moffitt is a part of NCDB and patients in the study by Zargar-Shoshtari et al are already covered in earlier studies. Similarly most patients in RICE dataset (Martinez-Chanza et al) are covered by NCDB dataset.
This introduces a big bias in any statistical analysis. Apart from the NCDB dataset the only study which has independent patients not covered by NCDB is by Kassouf et al which has only 11 patients. This makes any such systematic review or meta-analysis totally inappropriate.
Author Response
A detailed report on the amendments is presented below (each single change with appropriate citation of line number).
We totally agree with the Reviewer that our study is not free from biases resulting from potential overlap of some patients in selected articles (NCDB database). However, there are several differences in data coverage (duration, oncological outcomes). Thus, even with unavoidable biases, we ought to systematically present all evidences from currently available publications. Additional information regarding above mentioned limitations is provided in discussion section (Line 280-281).

Reviewer 2 Report
This study assessed the utility of adjuvant chemotherapy (AC) for patients with residual pT3-4 bladder cancer at cystectomy after NAC, or pN+ disease. They performed a meta-analysis using 6 studies and found that AC was associated with better OS versus surveillance for patients with pT3-4 disease, but not for those with pN+ disease.
The paper certainly addresses an important question for how to manage the many patients who do not have an excellent response to NAC. The findings that AC have a positive impact in those with pT3-4 disease are somewhat counter-intuitive, particularly since these patients already did not have a prior response. However, as the authors mention in limitations, this may be due to small sample size and retrospective nature of included studies.
One edit would improve the quality of the paper:
- Can the authors add to results as well as Table 2, the proportion of patients with variant histology in each of the included trials?
Author Response
A detailed report on the amendments is presented below (each single change with appropriate citation of line number).
“Can the authors add to results as well as Table 2, the proportion of patients with variant histology in each of the included trials?”
Our response: Data regarding variant histology were additionally provided (Line 170 - 171; Table 2).
Reviewer 3 Report
In this paper, the authors perform a systematic review and meta-analysis to assess differences in OS and DSS among patients with bladder cancer who had neoadjuvant chemotherapy and cystectomy for pT3-T4/N+ disease followed by either adjuvant chemotherapy (AC) or observation. They included 6 studies and 3096 patients, of whom only 741 received AC. Adjuvant chemo was associated with improved OS (HR 0.84, 95%CI 0.75-0.94) and DSS (HR 0.56, 95%CI 0.32-0.99) in the entire cohort, but subgroup analysis did not show a benefit for AC in patients with pN+ disease (HR 0.89, 95% CI 0.58-1.35).
The paper is very well written and well done. Though there are natural limitations to such a meta-analysis, the results are hypothesis generating enough to merit publication. I have a few small revisions that may improve the manuscript.
Intro:
- NAC provides a 5% absolute improvement in OS at 5 years (PMID 15939524). The reference you cited (#2) studies patients w complete response.
Methods:
- The abstract mentions inclusion of patients with cT2-T4aN0 disease, but that is not mentioned in the methods or anywhere else. Did you stratify by clinical stage or use it as inclusionary criteria?
- It would be nice to see the clinical stage and the actual chemotherapy regimens used as perhaps patients who received non-standard regimens were the ones who received AC?
- There are several other important confounders not mentioned i.e non-cisplatin based neoadj regimens, clinical stage, smoking status…
- A sensitivity analysis excluding unadjusted point estimates would be interesting
Results:
- Table 2: most studies did not specify type of chemo used or the # of cycles received, this needs to be addressed in limitations as does the lack of other important confounders as mentioned above
Discussion
- Nicely written but would mention the other major reason to consider CPI – most patients are not candidates for cisplatin based chemo after cystectomy
- Expand limitations to discuss poor recording of confounders as stated above. The receipt of “adequate” neoadjuvant chemo or adjuvant chemo is key here. Doesn’t seem like you can conclusively say that was the case in the studies included here.
Author Response
A detailed report on the amendments is presented below (each single change with appropriate citation of line number).
“NAC provides a 5% absolute improvement in OS at 5 years (PMID 15939524). The reference you cited (#2) studies patients w complete response.”
Our response: This citation was placed mistakenly and resulted from selection of wrong article via Mendeley plug in Microsoft Word. The previous citation was removed and the correct reference was cited (Line 297-300).
“The abstract mentions inclusion of patients with cT2-T4aN0 disease, but that is not mentioned in the methods or anywhere else. Did you stratify by clinical stage or use it as inclusionary criteria?”
Our response: We mentioned in both abstract and introduction section that cisplatin-based NAC followed by RC is the standard treatment of cT2-4a cN0 cM0 MIBC and 50% of patients receiving such treatment are eventually found to have advanced residual disease confirmed histopathologically (pT3–T4 and/or pN+) after RC. As it was stated in materials and methods section (inclusion criteria subsection), we included only studies comparing patients with pT3–T4 and/or pN+ disease at RC who received NAC and surgery followed by AC, with those without AC (Line 97-98).
“It would be nice to see the clinical stage and the actual chemotherapy regimens used as perhaps patients who received non-standard regimens were the ones who received AC?”
Our response: As suggested by the Reviewer, data regarding clinical stage was additionally provided in Table 2. Unfortunately, data regarding chemotherapy regimens in particular subgroups of patients were not reported in included studies. Thus, it was not possible to include them in the manuscript.
“There are several other important confounders not mentioned i.e non-cisplatin based neoadjuvant regimens, clinical stage, smoking status”
Our response: We focused mainly on major confounders reported in selected papers, used as adjustments in multivariable analyses determining the impact of AC (compared to surveillance) on survival parameters after NAC and RC for pT3/4 and/or pN+ patients. Clinical stage or smoking status was not used as adjusting factor in any selected article. We totally agree that detailed regimen types of NAC and AC are very important confounders, however, such data in selected studies were lacking.
“A sensitivity analysis excluding unadjusted point estimates would be interesting”
Our response: We initially performed sensitivity analysis and found no differences in HRs after excluding either individual studies or unadjusted point estimates. However, we decided to present only subgroup analysis in order to avoid excessive number of statistical data (without additional statistical significance). However, if the Reviewer finds it necessary, we can add these supplementary analyses to the manuscript.
“Table 2: most studies did not specify type of chemo used or the # of cycles received, this needs to be addressed in limitations as does the lack of other important confounders as mentioned above”
Our response: These issues were additionally addressed in the discussion section (Line 281-283).
“Nicely written but would mention the other major reason to consider CPI – most patients are not candidates for cisplatin based chemo after cystectomy”
Our response: It was additionally stated in the discussion section (Line 258-261).
“Expand limitations to discuss poor recording of confounders as stated above. The receipt of “adequate” neoadjuvant chemo or adjuvant chemo is key here. Doesn’t seem like you can conclusively say that was the case in the studies included here”
Our response: Limitations were expanded as suggested (Line 281-283).
Round 2
Reviewer 1 Report
It was my pleasure reviewing "Impact of Adjuvant Chemotherapy on Survival of Patients with Advanced Residual Disease at Radical Cystectomy following Neoadjuvant Chemotherapy: Systematic Review and Meta-Analysis" by Krajewski and colleagues.
As I mentioned previously this is not a meta-analysis as 5 studies are looking into the same dataset and have a significant overlap. The sixth study only ahs 11 patients.
The authors replied that there is overlap of some patients. This is wrong as all patients from smaller studies will be overlapping. This is the same NCDB dataset. This is not how a meta-analysis is done or should be done. An analogy is that if a phase III study gets published 4 times with longer and longer follow-up results we do not include the study 4 times in meta-analysis as they have the same set of patients.
Author Response
A detailed report on the amendments is presented below.
Our response: We can not totally agree with the Reviewer. As we stated in the study limitations, there is an inevitable risk of some patients’ overlap. However, we want to emphasise that most studies are covering different time intervals, and smaller studies include even broader time intervals than larger studies - thus this is not true that all patients will be overlapping. Moreover, some articles provided different oncological outcomes (e.g. study by Zargar-Shoshtari et al. reports only DSS), which are not included in another papers. Thus, we think that performing such quantitative and qualitative synthesis of available data is not a mistake. As an example, in the most recent study (Martinez-Chanza et al. published in European Urology Oncology) authors stated that: “there have been conflicting reports from retrospective series that cumulatively encompass >700 patients [Kassouf et al., Zargar-Shoshtari et al., Parker et al., Seisen et al., Sui et al.]”. In summary, we do not deny the possible bias resulting from patients’ overlap, but we can not say that included studies are one and the same.
Sincerely,
Authors